# Effects of Probiotics on Gut Microbiota: An Overview

**DOI:** 10.3390/ijms25116022

**Published:** 2024-05-30

**Authors:** Preethi Chandrasekaran, Sabine Weiskirchen, Ralf Weiskirchen

**Affiliations:** 1UT Southwestern Medical Center Dallas, 5323 Harry Hines Blvd. ND10.504, Dallas, TX 75390-9014, USA; 2Institute of Molecular Pathobiochemistry, Experimental Gene Therapy and Clinical Chemistry (IFMPEGKC), Rheinisch-Westfälische Technische Hochschule (RWTH) University Hospital Aachen, D-52074 Aachen, Germany; sweiskirchen@ukaachen.de

**Keywords:** probiotics, gut microbiota, dysbiosis, host immunity, intestinal flora, gut microbiome, beneficial bacteria, live microorganisms, intestinal homeostasis, gut barrier

## Abstract

The role of probiotics in regulating intestinal flora to enhance host immunity has recently received widespread attention. Altering the human gut microbiota may increase the predisposition to several disease phenotypes such as gut inflammation and metabolic disorders. The intestinal microbiota converts dietary nutrients into metabolites that serve as biologically active molecules in modulating regulatory functions in the host. Probiotics, which are active microorganisms, play a versatile role in restoring the composition of the gut microbiota, helping to improve host immunity and prevent intestinal disease phenotypes. This comprehensive review provides firsthand information on the gut microbiota and their influence on human health, the dietary effects of diet on the gut microbiota, and how probiotics alter the composition and function of the human gut microbiota, along with their corresponding effects on host immunity in building a healthy intestine. We also discuss the implications of probiotics in some of the most important human diseases. In summary, probiotics play a significant role in regulating the gut microbiota, boosting overall immunity, increasing the abundance of beneficial bacteria, and helping ameliorate the symptoms of multiple diseases.

## 1. Introduction

Recent evidence suggests that probiotics play a critical role in altering the composition of the gut microbiota and helping inhibit the colonization of pathogenic bacteria in the gut, thereby assisting the host in building healthy intestinal mucosa [1]. Probiotics primarily reside in the human intestine and work to balance the intestinal microbes. Common active bacterial preparations or probiotics include *Lactobacillus* or *Bifidobacterium* [2]. Probiotics have various roles such as boosting immunity, possessing anti-cancer properties, acting as anti-obesity agents, and having anti-diabetic effects in the treatment of chronic inflammatory and metabolic disorders [2]. They are dietary factors that exert regulatory effects on the structure and composition of the gut microbiome, primarily influencing host immunity.

The importance of probiotics was brought to light in 2001 when the definition of probiotics was first established. It was redefined as “live microorganisms that, when administered in sufficient amounts, provide numerous health benefits to the host” [3]. This definition was further refined in 2014 by a panel of scientists from the International Scientific Association for Probiotics and Prebiotics (ISAPP) as “this definition encompasses a wide range of microbes and applications, while capturing the essence of probiotics” [4].

The two dynamic functions of the gut microbiota include the modulation of the internal host environment and the influencing of the host immune response [5]. For example, intestinal microorganisms produce short-chain fatty acids (SCFAs), which enhance epithelial barrier function and reduce inflammation [5]. Another striking role of probiotics is in the innate and adaptive immunity of the human immune system. For instance, flagellin elicits an adaptive immune response and regulates the production of flagella by the microbiota. This, in turn, helps maintain the mucosal barrier and balance [6]. In a nutshell, the intestinal flora in the human body resists pathogens and maintains the integrity of the mucosal barrier.

The beneficial effects of probiotics are attributed to colonization resistance, organic acids, SCFA production, the competitive exclusion of pathogens, the normalization of altered microbiota abundance, the regulation of intestinal transit, direct antagonism, gut barrier reinforcement, and the neutralization of carcinogens [7]. Gut dysbiosis leads to several metabolic, intestinal, and cardiovascular conditions, and probiotics have shown promising effects as supplements or adjunct therapy in mitigating the effects of these diseases [8]. In this comprehensive review, we have summarized recent animal and human probiotic studies as well as the beneficial effects of probiotics in enhancing host immunity and modulating the gut microbiota. We have also highlighted the evidence-based health-promoting effects of probiotics.

## 2. The Human Gut Microbiota

The gut microbiota is a complex community of millions of microbes in the human colon, whose metabolic activity is extremely important in maintaining host homeostasis. It consists of several strains of bacteria and yeasts. The association between the gut flora and humans is commensal or mutualistic. Different sections of the gastrointestinal tract have different microbial compositions, with the colon harboring the highest microbiome population, while only a few species of bacteria are present in the stomach and small intestine. In total, 99% of the bacteria present in the gut are anaerobes [9].

Human microbiome projects have identified 2172 species of microbiome isolated from human beings classified into 12 different phyla [9]. The dominant strains of bacteria in the human gut belong to five major phyla: *Firmicutes*, *Bacteroidetes*, *Actinobacteria*, *Proteobacteria*, and *Verrucomicrobia* [10]. Alterations in its composition can cause a metabolic shift, changing the host phenotype [9]. The gut microbiota is influenced by environmental stimuli such as diet, the use of antibiotics or other medications, in addition to host factors like stress [11]. Important classes of microbiota-derived metabolites include SCFAs, bile acids, amino acids, trimethylamine N-oxide, tryptophan, and indole derivatives [12].

In the early stages of development, the diversity of microbiota is low. By around 2.5 years of age, the composition, diversity, and functional capability of infant microbiota resembles that of the adult microbiota. During adulthood, the composition of microbiota is relatively stable, although it can be altered by life events. However, at the age of 65, the composition of their microbiota and the abundance of several microbiota shifts. There is an abundance of *Bacteroidetes* phyla and *Clostridium* cluster IV, in contrast to younger persons, where cluster XIVa is more prevalent [10]. Given the close symbiotic relationship between the gut microbiota and the host, it is not surprising to find an altered gut microbiota in several diseases. The commensal microbiota load in the human gastrointestinal tract, measured in colony-forming units (cfu)/mL, gradually increases from the stomach to the jejunum to the colon (Figure 1).

Moreover, the alteration of beneficial gut microbiota can provoke gastrointestinal diseases such as irritable bowel syndrome, inflammatory bowel disease (IBD), celiac disease, colorectal cancer, and many other diseases. Importantly, an imbalance of the gut microbial communities and their metabolites has also been closely related to the dysfunction or illnesses of other organs (Figure 2).

## 3. Effects of Probiotics on Intestinal Homeostasis

It is crucial to maintain a dynamic balance of intestinal microbiota homeostasis in a stable ecological niche [15]. Probiotics increase the number of beneficial bacteria in the intestine through their own growth by promoting the growth of endogenous desirable microbial populations [15]. The second way in which probiotics regulate intestinal homeostasis is through competitive exclusion, which is a natural phenomenon of competition for nutrients and ecological niches. This process enhances the colonization of beneficial bacteria and prevents the growing of pathogenic bacteria [16]. For example, an in vitro study identified different carbohydrate-binding specificities of probiotic strains, such as *Lactobacillus rhamnosus*, *Lactobacillus mucosae* 1, and *Lactobacillus johnsonii BF01.* This study demonstrated their differential cell adhesion capabilities, enabling them to use a wide range of host cell receptor sites [17].

Another study found that a probiotic mix including the bacteria *Akkermansia muciniphila* and *Clostridioides difficile* occupies the ecological niche and acts as competitors by crowding out the pathogenic bacteria through mucopolysaccharide productivity [18]. In the mechanism of nutritional competition, bacterial biopolymers generated by *Bifidobacterium* and *Lactobacillus* are used as carbon sources by the gut microbiome to antagonize harmful bacteria and maintain a stable and shaped ecological environment [19]. Remarkably, probiotic intake has been associated with the adaptation of the physiological gut microbiome for carbon source competition via single nucleotide polymorphism [20]. Litvalk and his colleagues in 2019 found that in two model organisms, chickens and mice, *Salmonella* has a competitive advantage due to the increased oxygenation of the intestinal epithelium, while commensal enterobacteria protect the host by competing for oxygen. This prevents and reduces colonization by opportunistic pathogens [21]. Another mechanism of probiotics is regulating intestinal homeostasis through the secretion of metabolites. Intestinal *Lactobacillus* stimulates lactic acid production, activating hypoxia-inducible (HIF)-2α-mediated signaling, which is able to improve gut health [22]. This is evidenced by a significant reduction in *Vibrio cholerae* through probiotic administration in infant mice via lactic acid production [23]. In premature infants, *Bifidobacterium bifidum* and *Lactobacillus acidophilus* supplementation increases fecal acetate and lactate, while lowering the intestinal pH results in the accumulation of opportunistic gut pathogens like *Klebsiella*, *Escherichia*, and *Enterobacter* [24]. An interesting experiment showed that *Escherichia coli Nissle 1917 ECN* inhibits biofilm formation and disperses mature biofilm in pseudomonas aeruginosa to inhibit enterohemorrhagic *Escherichia coli* (EHEC). Probiotic *Escherichia coli* outcompetes pathogenic biofilms during dual-species biofilm formation [25]. In addition, the secretion of organic acids such as butyric acid, acetic acid, and propionic acid by probiotics is yet another important feature associated with their inhibitory activity against pathogens. Organic acids decrease pH conferring inhibitory activities on pathogens [26]. In summary, the competitive exclusion and secretion of metabolites such as lactic acid are a few of the underlying mechanisms involved in the regulation of intestinal homeostasis by probiotics.

## 4. Maintenance of Intestinal Epithelial Barrier by Probiotics

Intestinal epithelial cells act as the mediators of both the external and internal intestinal environments, working in conjunction with tight junctions (TJs) to form the mechanical intestinal barrier. In addition to the mechanical barrier, the intestinal barrier also includes a chemical barrier primarily composed of the mucus layer, with an immune barrier present as well. Disruptions or alterations to the intestinal mucus layer can increase the risk of conditions like leaky gut syndrome and IBD [27]. The single layer of intestinal epithelial cells is organized into villi and crypts [28]. Pathogens must first penetrate the mucosal barrier before reaching the epithelium. Changes in the content and structure of mucus can impact the barrier function, as harmful gut microbes can degrade the mucus [16]. An important study by Krndija et al. in 2019 found that the homeostatic renewal of adult gut epithelium is supported by the active migratory forces of dually polarized epithelial cells, which are based on actin-rich basal protrusions moving in a specific direction [29].

Multiple studies suggest that probiotics protect the barrier function by promoting mucus secretion. For example, *Lactobacillus plantarum BMCM12* secretes extracellular proteins that weaken pathogen adhesion and protect the intestinal barrier [30]. The integrity of the epithelial barrier is also enhanced by probiotics. For instance, the probiotic mixture VSL#3 or *Lactobacillus rhamnosus GG* maintains the integrity of the epithelial barrier in mice [26]. Remarkably, the metabolites of probiotics such as butyric acid maintain barrier integrity by enhancing oxygen consumption in the epithelium, subsequently increasing the expression of barrier-protective HIF-responsive genes [30]. Another interesting finding is that probiotics reduce intestinal permeability, helping to protect the human gut. This was reported by Caballero France wherein probiotics prompt goblet cells to produce mucin, inhibiting the adherence of pathogens [16].

One study showed that the *Limosilactobacillus reuteri D8* strain stimulates the regeneration of intestinal stem cells by increasing the number of Paneth cells in the crypts, promoting the repair of intestinal mucosa [31]. In a similar context, administering *Akkermansia muciniphila* for 4 weeks accelerated the proliferation of Lgr5^+^ intestinal stem cells and promoted the differentiation of Paneth cells and goblet cells in the small intestine [32]. Another study found that the ingestion of *Lactobacillus reuteri* ATCC PTA 4659 could maintain the integrity of the intestinal barrier and help to protect the mucosal layer by enhancing the expression of epithelial heat shock proteins 25 (HSP25) and 70 (HSP70), thereby strengthening epithelial cell structural protein interactions [33]. Furthermore, the supplementation of *Bifidobacterim adolescentis IVS-1* and *Bifidobacterium lactis BB-12* to obese patients increased the total *Bifidobacteria* and strengthened gut barrier function [34].

Another mechanism of probiotic-mediated barrier regulation is by enhancing the action of TJ proteins on epithelial cells [35]. In this context, lipopolysaccharide (LPS), representing an intestinal-derived bacterial toxin, consisting of a lipid and a polysaccharide, could activate Toll-like receptor 4 (TLR4) and downstream NF-κB signaling pathway. This causes a decrease in TJ protein expression and an increased likelihood of LPS translocation. This can cause a localized and systemic inflammation through cytokine production. A probiotic formulation, combined with dexamethasone, enhances intestinal barrier integrity by blocking LPS translocation and inhibiting the TLR4/NF-κB pathway, alleviating symptoms in an autoimmune hepatitis mouse model [36]. The increase in taurine levels by probiotics can also trigger the expression of TJ proteins, which in turn can reduce intestinal permeability and inhibit gut leakage [37].

A randomized controlled trial in dogs reported that a compound probiotic powder could improve the homeostasis of the intestinal mucosa by regulating the expression of TJ proteins, E-cadherin, and occludin [38]. Probiotics protect the mucosal structure by promoting mucin secretion. For example, *Akkermansia muciniphila* releases one or more active metabolites that enter epithelial cells [39]. *Bacteroides thetaiotaomicron* has been shown to increase the intestinal mucus layer thickness by promoting mucin 2 (MUC2) production, restoring the intestinal mucosal barrier, and reducing LPS translocation [40]. It is interesting to note that odorant-binding proteins influence the microbiota and host physiology in patients with IBD [41].

Probiotics modulate the intestinal barrier by influencing mucosal barrier-associated effector immune cells, such as lymphocytes. *Lactobacillus plantarum G83*, which was isolated from giant panda feces, demonstrated protection against enterotoxigenic *Escherichia coli K88* infection in a mouse model [42]. Interestingly, commensal fungi like *Candida albicans* or *Saccharomyces cerevisiae* offer protection to the host against mucosal injury, thereby enhancing the circulation of immune cells [43]. In patients with NAFLD, probiotic supplementation has been shown to increase CD8^+^ cells, which helps regulate intestinal mucosal immune functions and reduce intestinal permeability [44].

In a study involving weaned piglets, the use of *Saccharomyces cerevisiae* improved gut health by enhancing mucosal secretory immunoglobulin A secretion, antioxidant capacity, and intestinal immunity, thereby reducing pathogen colonization [45]. In summary, probiotics have been shown to enhance gut mucosal integrity by protecting mechanical, chemical, and immune barriers, ultimately reducing inflammation, leaky gut, and pathogen translocation. Figure 3 illustrates the beneficial effects of probiotics on intestinal barrier function and homeostasis.

## 5. Immunoregulatory Effects of Probiotics

The versatile role of probiotics extends to enhancing the intestinal mucosal immune defense system, reducing the risk of pathogen penetration through the intestinal epithelium [46]. Probiotics contribute to the host’s innate and adaptive immunity by promoting T cell differentiation, modulating cytokine levels, and increasing IgA^+^ cells [47]. These effector components establish a signaling network between different types of immune cells. In animal models, *Bifidobacterium longum* and *Bifidobacterium infantis* have been shown to increase interleukin (IL)-10, while inhibiting the levels of IL-12, IL-17, and IL-23 through increased FOXP3 lymphocyte expression and stimulating T cell differentiation [48]. Notably, in certain allergies, probiotics have been shown to regulate the balance of T cell subsets by promoting Th2 to Th1 conversion, thereby reducing allergy symptoms [49].

The innate lymphoid cells regulate epithelial cell interaction, the gut microbiota, and the adaptive immune system. Group 3 lymphoid cells are enriched in mucosal tissues. They produce IL-22 and IL17A [50]. Increasing the gut microbial diversity is another mode of action by which probiotics regulate the host immune system. For example, in a colitis mouse model, probiotics could increase microbial diversity, modulate the levels of inflammatory factors, and alleviate symptoms [51]. Notably, several clinical studies have demonstrated enhanced efficacy of immunotherapy in cancers in combination with probiotics. In a cancer mouse model, *Lactobacillus rhamnosus* probiotic M9 (5 × 10^9^ cfu/day) effectively restored the mouse gut microbiota, synergistically improving the anti-tumor effect of anti-PD1 therapy and mouse survival [52]. A melanoma model demonstrated that *Lactobacillus kefiranofaciens ZW18* (1 × 10^9^ cfu/day) had the best anti-melanoma effect when administered along with PD1-inhibitor anti-cancer therapy, enhancing the body’s immune response by promoting the infiltration of CD8^+^ T cells [53].

The chemical and physical barrier of the intestine keeps the gut microbiota and immune cells separate, preventing abnormal immune reactions and allowing them to co-exist in harmony within the host [35]. The disruption of mucins, TJ proteins, and goblet or Paneth cells in the intestinal barrier can trigger autoimmune diseases and inflammation [54]. Probiotics such as *Escherichia coli Nissle 1917*, *Lactobacillus plantarum ZLP001,* and a few *Lactobacillus reuteri* strains regulate these proteins, thereby regulating gut immunity. For example, the probiotic mixture VSL#3 that consists of eight live bacterial strains, restored intestinal mucosal TJ protein damage and increased the expression of mucin (MUC)2, MUC3, and MUC5AC in several human colon cancer cell lines [55]. Some *Lactobacillus* strains can lower the inflammatory responses by inhibiting the NF-κB pathway or the phosphorylation of MAPKs [56]. Gut microbial metabolites such as SCFAs, amino acids, bile acids, and vitamins involved in immune regulation are modulated by probiotics and influence the outcome of intestinal inflammation. SCFAs exert anti-inflammatory effects by binding to specific receptors on epithelial cells, inhibiting the production of pro-inflammatory cytokines [57]. Probiotic metabolites such as amino acids and amino acid derivatives interact with immune cell surface receptors and exert anti-inflammatory effects [57]. An interesting effect of probiotics is their metabolization of primary bile acids to secondary bile acids which bind to various GPCR—G-protein coupled receptors or nuclear receptors to control mucosal immune activity and decrease inflammation [58].

Probiotics have been shown to induce the IgA cycle and stimulate the maturation of the humoral immune system [59]. They can also boost the population of macrophages and dendritic cells, which play a crucial role in the immune system by identifying and eliminating pathogens. Dendritic cells can migrate to mucosa-associated lymphoid tissue or drain into lymph nodes through the antigen barrier [60]. In one study, it was found that certain *Lactobacillus* probiotic strains triggered an inflammatory response in macrophages in vitro by producing cytokines, regulating reactive oxygen species formation, and activating the TLR2 pathway [61].

The germ-free mouse models enable the understanding of the relationship between the gut microbiota and the immune system of the host. One experiment reported an absent mucus layer and altered IgA secretion in germ-free animals [28]. In brief, probiotics influence the host’s immune system by interfering with immune cells and inflammatory factors, regulating the gut microbiome and its metabolites, and repairing the gut barrier. The immunomodulatory functions of probiotics are illustrated in Figure 4.

## 6. Probiotic Regulation of Signaling Molecules Secretion

It is important to gain a deeper understanding of the metabolites produced by the gut microbiota, which serve as signaling molecules in host–gut microbiota interactions [64]. SCFAs act as histone deacetylase inhibitors, preventing excessive histone acetylation [65]. SCFAs interact with G-protein coupled receptors (GPCRs), as demonstrated by the interaction of SCFAs produced by the gut microbiota of pregnant mice with GPR41 on the sympathetic nerve of the embryo and GPR43 on the pancreas. This promotes the differentiation of nerve cells and islet β-cells, shaping the embryo’s metabolic system development [66]. Several SCFA-producing probiotics help alleviate symptoms in metabolic disorders, IBD, and cancers. For example, probiotics have been shown to control blood glucose levels by influencing gut microbes and SCFA production [67].

One animal study showed that administering *Lactococcus lactis* G15 and Q14 enhanced epithelial barrier function, improved glucose tolerance, and decreased lipid levels by activating SCFA-producing bacteria through the G-protein coupled receptor 43 (GPR43) pathway in type 2 diabetes mice [68]. In an ulcerative colitis model, supplementing live *L. acidophilus* increased SCFA levels and activated GPCRs to inhibit NLRP3 inflammation and facilitate autophagy [69].

The dysregulation of the tryptophan–kynurenine pathway is associated with neurological disorders and autoimmunity. Probiotic supplementation can help mitigate abnormal tryptophan–kynurenine pathway metabolism. For example, circulating kynurenine levels were increased in mice under constant stress, and *Lactobacillus* helped reduce these levels by blocking intestinal indoleamine 2,3-dioxygenase (IDO1) expression that is the main enzyme responsible for the conversion of tryptophan to kynurenine and inhibiting kynurenine metabolism [70]. In a similar context, *Lactobacillus plantarum 299V* improved cognitive performance and enhanced SSR1 treatment in patients with depressive disorders by decreasing the concentration of kynurenine [71].

Table 1 summarizes the metabolites produced by probiotics alone or in interaction with the gut microbiota and their effects.

## 7. Human and Animal Trials on Probiotics

Several animal and human studies have proven the importance of probiotic intervention in various diseases and its beneficial effects. Diabetes mellitus is the most common metabolic disorder posing the greatest challenge to the healthcare system worldwide, causing life-threatening complications [86,87]. Probiotics have been proven beneficial in alleviating the symptoms of insulin resistance. A few of the important and interesting animal and human studies on probiotics have been discussed here. A remarkable study by Hou and his colleagues demonstrated that probiotic intervention in healthy adults from six Asian regions increased the abundance of *Lactobacillus* spp., along with other beneficial bacteria such as *Roseburia*, *Coprococcus*, and *Eubacterium rectale*, and inhibited harmful bacteria like *Blautia* and *Ralstonia* [88]. Another study reported that long-travel-associated gut dysbiosis can be prevented by the administration of probiotics, as evidenced by an increase in beneficial microorganisms and the inhibition of harmful microbes such as *Klebsiella pneumoniae* and *Clostridium leptum* [89].

In neonatal rats, *Bacteroides* and *Bacillus* restored the intestinal epithelial barrier function and inhibited enterocolitis [90]. Piewngam and his colleagues reported that the oral intake of *Bacillus subtilis* spores inhibits the fecal streptococci regulator activity of *Enterococcus faecalis*, thereby preventing bacteremia caused by the translocation of this Gram-positive commensal bacterium [91]. Another recent prospective study showed that several bacteria belonging to the *Lactobacilli* genus interact with intestinal commensal bacteria, thereby reducing the colonization and growth of *Enterobacteriaceae* in the intestine [92]. Strikingly, Huang and colleagues reported that spraying the probiotic fermented liquid (*Lacticaseibacillus casei*, *Lactiplantibacillus plantarum*, and *Lactobacillus rhamnosus* probio-M9) into the living environment of piglets significantly improved their growth and immunity and decreased the abundance of *Escherichia coli* [93]. Furthermore, ingesting *Limsilactobacillus reuteri DSM 17648* along with standard triple antibiotic therapy increased the eradication rate of *Helipbacter pylori*, alleviating gastrointestinal discomfort by regulating the gut microbiota [94].

An interesting study revealed that injecting probiotics into obese animals reversed gut dysbiosis and inflammatory response in mice [95]. Another trial demonstrated that patients with Crohn’s disease or ulcerative colitis, collectively referred to as IBD, experienced an increase in probiotic bacteria such as *Bifidobacterium* and *Lactobacillus* when consuming yogurt. This resulted in improved intestinal function [96]. A randomized controlled trial showed that live *Lactobacillus plantarum 299V* could reduce intestinal permeability and suppress inflammatory response in patients with obstructive jaundice [97]. Another intriguing clinical trial found that 12 weeks of *Bifidobacterium longum* therapy significantly reduced the expression of several pro-inflammatory cytokines in patients with IBD, including IL-6, IL-8, and TNF-α [98]. The immunomodulatory effect of microbial metabolites derived from probiotics was evidenced by a decrease in inflammatory response caused by renal macrophages, a decrease in gut microbiota imbalance, and an increase in beneficial metabolites protecting the kidney in a double renal ischemia–reperfusion mouse model after *Lactobacillus casei* Zhang supplementation with 1 × 10^9^ cfu per day [99].

In diet-induced obese mouse models, the administration of the probiotics *Bifidobacterium animalis* and *Lactobacillus paracasei* for 12 weeks, along with prebiotics, significantly alleviated diet-induced metabolic and immunity disorders [100]. An interesting randomized controlled trial showed that *Lactobacillus acidophilus LA-5* and BB-12 at 1 × 10^9^ cfu/day for six weeks improved glucose tolerance in type 2 diabetic subjects, with an increase in acetic acid and a decrease in TNF-α and Resistin [101]. Several clinical trials have demonstrated that probiotics can decrease blood glucose levels, glycated hemoglobin, and diabetes symptoms in patients with type 2 diabetes mellitus and gestational diabetes mellitus [102,103]. Similarly, insulin secretion was improved in a diabetic mouse model after the administration of probiotics by triggering glucagon-like peptide 1 (GLP-1) and GPR43/41 expression [104]. Additionally, administering a *Bifidobacterium* strain has been shown to alleviate metabolic syndrome, and *Bifidobacterium animalis* ssp. *lactis GCL2505* decreased visceral fat and increased glucose tolerance [105]. A significant finding reported that *Propionibacterium freudenreichii* with probiotic potential inhibits colorectal cancer proliferation and promotes the apoptosis of cancer cells by producing acetate and propionate [106]. A groundbreaking discovery on *Clostridium butyricum* suggests it could relieve Parkinson’s disease by increasing GPR41/GPR43 levels and GLP-1 receptors in the brain in mouse models [107]. A probiotic strain of *Lactobacillus gasseri* exerts anti-inflammatory effects in mouse colitis models, maintaining the integrity of the gut barrier [108]. Table 2 summarizes several important randomized controlled trials on the effect of probiotics in type 2 diabetes mellitus.

## 8. Probiotics and Obesity

Obesity, a pandemic in recent times, is closely related to the disorders of the intestinal flora [114]. The intestinal flora has the capacity to prevent the local inflammation of adipose tissue by enhancing immunity and preventing adipose tissue inflammation. Factors contributing to the pathogenesis of fatty liver disease include adipose tissue dysfunction/inflammation, the dysbiosis of the gut microbiota, and gut barrier function regulating several intrahepatic metabolic and inflammatory pathways. Therefore, probiotics and prebiotics may play a therapeutic role in fatty liver diseases by modulating the gut microbiome [115]. One study reported that the genetic sequencing of fecal samples from obese patients showed significantly fewer *Bacteroidetes* and more *Firmicutes* compared with lean volunteers [116]. In an interesting study, normal cecal microbiota was introduced into adult germ-free mice and two weeks later, despite a reduction in food intake, adult germ-free mice had a 60% increase in insulin resistance and body fat [117].

## 9. Probiotics and Skin

The intestinal microbiota is closely related to skin diseases such as acne, psoriasis, and atopic dermatitis. For example, probiotics are very effective in treating atopic dermatitis by enhancing immunity. A study conducted in Norway reported that the incidence of atopic dermatitis can be effectively decreased by supplying probiotic milk to women and infants pre- and post-delivery [118]. Another interesting study showed that *Bifidobacterium* levels in the intestines of patients with atopic dermatitis were lower as levels were negatively correlated with the severity of the disease in patients with atopic dermatitis compared to healthy controls [119].

Another common skin disease is acne, which is characterized by changes in keratin, inflammation, hormone-induced hyper seborrhea, and decreased immunity [120]. Studies have reported that *Lactococcus* sp. *HY449* directly inhibits the occurrence of *Propionibacterium acnes* through the production of anti-microbial proteins [121]. Furthermore, consuming probiotic *Lactobacillus bulgaricus* tablets and *Lactobacillus acidophilus* has been shown to improve the condition of patients with acne [118]. An animal study demonstrated that an oral solution containing *Lactobacillus reuteri* significantly reduced the number of major histocompatibility cells around the hair follicles compared to controls [121].

Another skin condition, psoriasis, is a chronic inflammatory skin disease presenting as erythematous thick scaly plaques on the skin. The association of psoriasis and the gut microbiota can be evidenced by the severe dysregulation of the gut flora in patients with psoriasis, decreased diversity of certain taxa, and alterations in abundance [122]. Scher and colleagues demonstrated that the gut microbiota in patients with psoriasis was less diverse and that, further, the abundance of individual phyla is different in patients suffering from psoriatic arthritis and psoriasis of the skin [123].

Yan et al. showed that the prevalence of psoriasis was negatively associated with the concentration of *Actinomycetes* as assessed by psoriasis activity and severity index score [124]. The oral administration of *Lactobacillus pentosus GMNL-77* effectively treated skin inflammation in mice [125]. Figure 5 depicts the implications of probiotics in notable diseases.

## 10. Probiotics: Method of Delivery

Recently, the methods of probiotic delivery have gained attention. For example, *Bifidobacterium longum* encapsulated with artificial enzymes could enhance the colonization time of probiotics in the gut, enhancing its anti-inflammatory effect [128]. In a similar context, another study reported that probiotics encapsulated by prebiotics were specifically enriched around colon cancer lesions in mice, effectively inhibiting colon cancer [129]. Strikingly, live *Lactobacillus rhamnosus* encapsulated in nanoparticles can modulate the lung microbiota along with promoting host immune recovery [130]. Finally, the latest technologies such as single cell-omics, isotope tracking, and CRISPR/Cas technology will unravel new host delivery strategies and immunomodulatory effects of probiotics.

In addition, the response to probiotic supplementation is known to vary with gender and genetic differences. One study investigated the sex-dependent effects of probiotics on gut microbiota profiles and found a significant reduction in pro-inflammatory gut microbes in women compared to men after probiotic supplementation. Furthermore, peripheral immune cell profiling showed that probiotics decreased the proportions of dendritic cells and CD14 monocytes in men, but not in women. This indicates sex-specific responses in the regulation of the gut microbiota by probiotics [131]. Another study demonstrated a high proportion of *Firmicutes* in female athletes compared to males after probiotic supplementation [132]. An interesting study conducted at Gothenburg University reported differences in the clinical symptoms of *Salmonella* infection between male and female patients after *Lactobacillus plantarum* supplementation [133]. In summary, the response to probiotics varies with multiple factors such as gender and race. Moreover, these studies demonstrate that defining a probiotic can be complicated and may be highly challenging because there are many factors that might impact the biological activity of a specific “probiotic”.

## 11. Taxonomic and Metabolomic Profiling

Nowadays, it is well-accepted that the loss or abundance of specific taxonomic groups is associated with a number of disorders such as type 2 diabetes, obesity, IBD, and some types of cancers. Therefore, taxonomic profiling may provide important mechanistic insights relevant to understanding the initiation or progression of these diseases. Consequently, mathematical frameworks have been established for analyzing gut microbial communities in cohorts suffering from different diseases. For example, the global interspecies metabolic interaction network of the human gut microbiota, NJS16, has identified a community-scale infrastructure of metabolic influence with the type 2 diabetes mellitus gut ecosystem [134]. Other studies focus on the metabolites produced by the gut microbiome. In one landmark study, the authors combined integrative metagenomic and metabolomic information from the gut microbiome with optimization techniques from machine learning and proposed an ecology-based computational algorithm, GutCP, that identified high-consensus cross-feeding interactions between 72 prominent gut microbial species and 221 gut metabolites. This suggests that mathematical models and artificial intelligence have the potential to provide tractable information about the ecological inference in the gut microbiome [135].

However, machine learning methods and artificial intelligence in microbiome analysis still have some limitations and bottlenecks [136,137]. Nevertheless, the success of artificial intelligence, along with the evaluation of big data sets from defined cohorts, will pave the way for future applications and the development of gut microbiota-targeted strategies for the treatment and prevention of human diseases [138].

## 12. Conclusions

The modulation of the gut microbiota through reshaping host–microbiota interactions can be achieved with strategies such as probiotics and personalized nutrition as adjunctive therapy. Pathogenic microorganisms alter the homeostasis of the gut microbiota, leading to an increased risk of related diseases. Probiotics inhibit these pathogens by stimulating epithelial barrier function, secreting anti-microbial components, competitively excluding pathogens by binding sites, and limiting their access to nutrients. The diversity, variability, and complexity of the gut microbiota can be disrupted by multiple factors, causing various illnesses. Interventions with probiotics and their modulation of the gut microbiota help sustain good health and alleviate these diseases. Despite the beneficial effects of probiotics, there are variations in some trials that do not show beneficial effects. Therefore, intensive research could provide a deeper understanding of probiotics for practical and clinical applications. Progress in analyzing the gut microbiome through the adoption of systems biology approaches and the use of artificial intelligence will effectively promote an understanding of the impact of the microbial community and its metabolites in the pathogenesis of individual diseases. Undoubtedly, this will also be beneficial in developing novel microbe-targeted therapies for modifying the gut microbiota or its metabolites in personalized and precision medicine.

## Figures and Tables

**Figure 1 ijms-25-06022-f001:**
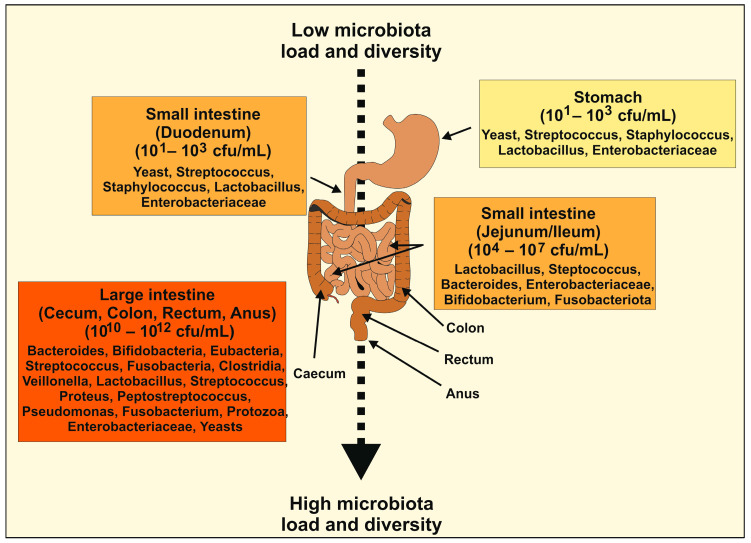
Microbiota load and diversity vary throughout the gastrointestinal tract. In a healthy individual, the microbiota count, measured as colony-forming units (cfu)/mL, in the human gastrointestinal tract increases from the stomach/duodenum to the jejunum/ileum to the colon. Additionally, the regional diversity in the gastrointestinal microbiome, known as the microbial landscape, increases from the mouth (rostral) to the anus (caudal). Importantly, there is significant intra- and interpersonal variation in the composition of the human microbiome, which is further influenced by various factors such as the mode of infant delivery and feeding, aging, diet composition, geography, medication, stress, and many others [13].

**Figure 2 ijms-25-06022-f002:**
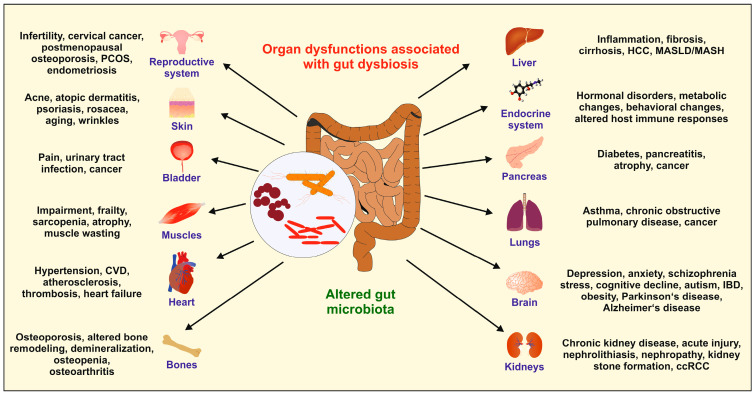
Organ dysfunction as a consequence of dysbiosis. The balance of the physiological microbiota community (eubiosis) in the gut is crucial to ensure an individual’s health. Factors such as an unhealthy diet, excessive fasting, alcohol consumption, smoking, physical or psychological stress, chronic inflammation, and the overuse of antibiotics can lead to dysbiosis, which can negatively impact the health of organs. Diseases associated with abnormalities and imbalances in the gut microbiota can vary greatly and affect all organs. The abbreviations used are as follows: ccRCC, clear cell renal cell carcinoma; CVD, cardiovascular disease; HCC, hepatocellular carcinoma; IBD, inflammatory bowel disease; MASH, metabolic-associated steatohepatitis; MASLD, metabolic dysfunction-associated steatotic liver disease; PCOS, polycystic ovary syndrome. For more information on the impact of the gut microbiota and its role in health and disease, please refer to [14].

**Figure 3 ijms-25-06022-f003:**
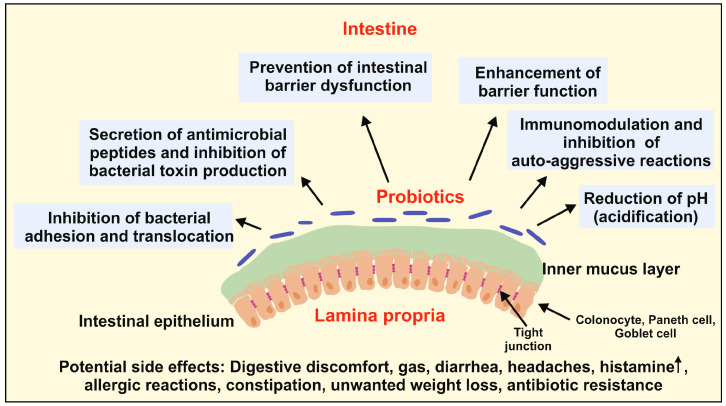
Beneficial effects of probiotics on the function and homeostasis of the intestinal barrier. Probiotics impact the intestinal barrier by preventing intestinal barrier dysfunction through enhancing the expression of tight junction proteins and inhibition of bacterial adhesion and translocation. Moreover, probiotics lower the luminal pH value through the secretion of anti-microbial active peptides, acetic and lactic acids, which inhibits the growth of non-commensal pathogens and the production of their bacterial toxins. They further modulate the host’s immune system, thereby inhibiting auto-aggressive reactions. However, probiotics can have the depicted side effects such as digestive discomfort, headaches, constipation, unwanted weight loss, and others. Please note that the image does not show the full structure of the gastrointestinal wall including its mucosa, submucosa, muscularis, and serosa.

**Figure 4 ijms-25-06022-f004:**
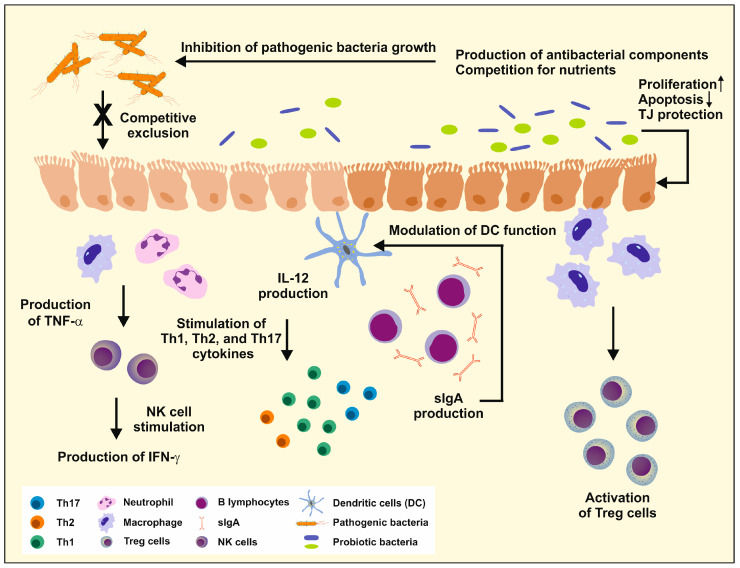
Probiotics are effective in strengthening the host’s immune system. They have various effects on an individual’s health, including inhibiting the growth of pathogenic bacteria through the production of antibacterial components and competition for nutrients. Probiotics also compete for mucosal adhesion sites and stimulate the expression of cytokines that affect cell proliferation and apoptosis. Additionally, they promote the production of tight junction (TJ) proteins and stimulate or modulate immune cell function in natural killer (NK) cells and dendritic cells (DC). The increased production of secretory IgA (sIgA) helps protect the intestinal epithelium from enteric pathogens and their toxins. More information on the impact of probiotics on immunity and their mechanism of action on immune cells can be found in other publications [62,63]. Please note that the gastrointestinal wall is only roughly depicted and does not show its fine structure, which includes the mucosa, submucosa, muscularis, and serosa.

**Figure 5 ijms-25-06022-f005:**
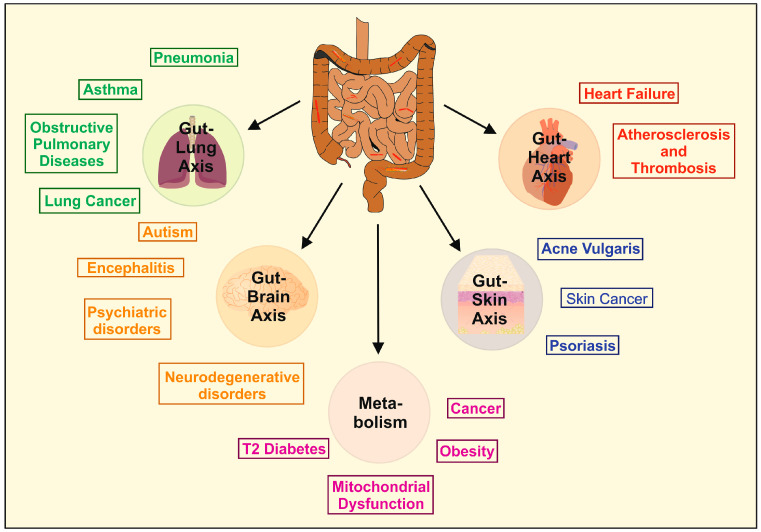
Applications of probiotics in the therapy of human diseases. The manipulation of the gut microbiota may be beneficial in treating diseases of the heart, skin, brain, and lung through what are known as interorganic axes (e.g., gut–heart axis, gut–skin axis, gut–brain axis, and gut–lung axis). The diseases listed are just a few examples where probiotics have shown benefits. Additionally, they have a positive impact on general metabolism, making them useful in the treatment of type 2 diabetes, obesity, and cancer, likely by addressing mitochondrial dysfunction. For more information on the relationship between microbiota, the gut, and organs, as well as potential treatment approaches for various pathologies, please see reference [126]. This figure has been adapted from the work of Gebrayel and colleagues [127].

**Table 1 ijms-25-06022-t001:** Sources of probiotic metabolites and their effects *.

Metabolites	Mechanism/Source of Production	Effects	References
SCFA-acetate	Diet and endogenous production through acetyl-CoA	Increased satiety, weight loss, improved insulin sensitivity, and decreased pro-inflammatory cytokines	[72,73,74]
SCFA-propionate	Dietary fiber fermentation	Decreased weight gain, intestinal and hepatic gluconeogenesis, and decreased pro-inflammatory cytokine levels	[75,76]
SCFA-butyrate	Dietary fiber fermentation	The maintenance of mucosal integrity, the regulation of local and systemic immunity, anti-obesity effects, the stimulation of leptin synthesis, and the release of anorexigenic hormones	[77,78,79]
TMAO	Egg, milk red meat, and fish	Increased levels are associated with adverse cardiovascular disease	[80]
Tryptophan metabolites, indole derivatives, and tryptamine	Gut microbiota-derived	Anti-microbial effects, anti-obesity properties, appetite suppression, and slow gastric emptying	[81,82]
Primary bile acids (e.g., cholic acid)	Liver	The facilitation of fat digestion and nutrient absorption, and protection of the mucosal barrier	[83,84]
Secondary bile acids (e.g., deoxycholic acid and lithocholic acid)	Produced in colon	The inhibition of *Clostridioides difficile* spore germination and associated colorectal cancer and HCC	[83,84]
Polyamines (e.g., putrescine, spermidine, and spermine)	Lower GI—synthesized by the gut microbiome Upper GI—food-derived	The regulation of stress, antioxidant effects, and impact on cell proliferation and differentiation	[85]

* Abbreviations used are as follows: GI, gastrointestinal tract; HCC, hepatocellular carcinoma; SCFA, short-chain fatty acid(s); TMAO, trimethylamine N-oxide.

**Table 2 ijms-25-06022-t002:** Selected randomized controlled trials on the effects of probiotics in therapy of type 2 diabetes mellitus *.

Intervention	Treatment	Duration of Intervention	Changes from Baseline	Reference
Probiotic capsules/placebo capsules (27/27)	*Lactobacillus acidophilus* (2 × 10^9^ cfu), *Lacticaseibacillus casei* (7 × 10^9^ cfu), *Lactobacillus rhamnosus* (1.5 × 10^9^ cfu), *Lactobacillus bulgaricus* (2 × 10^8^ cfu), *Bifidobacterium breve* (2 × 10^10^ cfu), *Bifidobacterium longum* (7 × 10^9^ cfu), and *Streptococcus thermophiles* (1.5 × 10^9^ cfu)	8 weeks	ΔFPG (mg/dL): 1.6 ± 6 ΔHbA1c (%): −0.3 ± 0.37	[109]
Synbiotic food/placebo food (62/62)	*Lactobacillus sporogenes* (1 × 10^7^ cfu)	6 weeks	ΔFPG (mg/dL): 22.3 ΔInsulin (µIU/mL): −1.75 ± 0.6	[110]
Probiotic capsules/placebo capsules (16/18)	*Twice weekly 1500 mg capsules containing Lactobacillus acidophilus*, *Lactobacillus bulgaricus*, *Lactobacillus bifidum,* and *Lacticaseibacillus casei*	6 weeks	FPG (mg/dL): 158.69 ± 16.38 vs. 158.56 ± 13.7 Insulin (ng/mL): 0.35 ± 0.11 vs. 0.41 ± 0.16	[111]
Synbiotic shake/placebo shake (10/10)	*Lactobacillus acidophilus* (2 × 10^10^ cfu), *Bifidobacterium bifidum* (2 × 10^10^ cfu)	30 days	FPG (mg/dL): 116.78 ± 18.96 vs. 191.11 ± 18.31	[112]
Synbiotic bread/control bread (30/30)	A total of 3 times a day in a 40 g package for a total of 120 g/day	8 weeks	ΔFPG: (mg/dL): 6.04 ± 8.41 ΔHbA1c (%): −0.28 ± 0.06 ΔInsulin (µIU/mL): −2.05 ± 1.03	[113]

* Abbreviations used are as follows: cfu, colony-forming unit(s); FPG, fasting plasma glucose; HbA1c, glycated hemoglobin.

## Data Availability

This review only presents data that were previously published. No new data were generated.

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
