# Peer review of "Effects of Probiotics on Gut Microbiota: An Overview"

_ijms, 2024, doi:10.3390/ijms25116022_

Round 1

Reviewer 1 Report

Comments and Suggestions for Authors

Chandrasekaran et al. summarize the multifaceted roles of probiotics in maintaining and improving gut health. It focuses on the interactions between probiotics and gut microbiota and their implications for human health. The authors outline the essential mechanisms through which probiotics modulate gut microbiota, such as competitive exclusion, secretion of beneficial metabolites, and maintenance of intestinal barrier function. They highlight the immunomodulatory effects of probiotics and provide a synopsis of various human and animal trials illustrating the therapeutic potential of probiotics in managing diseases like diabetes, obesity, and inflammatory bowel diseases.

Major issues:

1.     While the paper thoroughly reviews the benefits of probiotics, it does not adequately address the variability in individual responses to probiotics. It would be beneficial to discuss factors that contribute to this variability, such as genetic differences and existing gut microbiota composition.

2.     I suggest a better description of the relationships between gut microbes and metabolites in the third section “Effects of Probiotics on Intestinal Homeostasis”. Although the authors summarize some literature evidence of interactions between gut microbes and metabolites, some works have systematically captured their interactions that have been found in many pieces of literature (Akshit Goyal et al., Nature Communications 2021; Jaeyun Sung et al., Nature Communications 2017). Therefore, the authors need to properly summarize existing papers.

Minor issues:

1.     Line 26: “a healthy intestinal mucosa” -> “healthy intestinal mucosa”

2.     Line 77: “Once individual reach” -> “Once an individual reaches”

3.     Line 96: “ilnesses” -> “illnesses”

4.     Line 140: “inhibits the biofilm formation and disperse” -> “inhibits the biofilm formation and disperses”

5.     Line 209: “Probiotic supplementation in patient” -> “Probiotic supplementation in patients”

6.     Line 352: “suppress inflammatory response” -> “suppress the inflammatory response”

Comments on the Quality of English Language

Grammatical edits are suggested in my minor comments.

Author Response

Dear reviewer 1,

Thank you for taking the time to review our paper and for your thoughtful and constructive comments. Please find our response to your comments and suggestions in the attached pdf-file.

Regards

Ralf Weiskirchen

Reviewer 2 Report

Comments and Suggestions for Authors

The paper “Effects of Probiotics on Gut Microbiota: An Overview” provides firsthand information on gut microbiota and their influence on human health, the dietary effects of diet on gut microbiota, and how probiotics alter the composition and function of the human gut microbiota, along with their corresponding effects on host immunity in building a healthy intestine. Here are some questions that need to be further improved or explained.

Comments:

Q1. Authors are advised to be cautious about using statements that are too sure or confident,like “This comprehensive review is unique”, etc. The gut microbiota and human health benefits papers (including review paper) are not so absent.

Q2. The article introduces the types and reported functions of many probiotics. How do we determine which species are probiotics?

Q3. How do probiotics exert immunomodulatory effects? By direct contact or some specific metabolites that could enter into blood? Are there large numbers of immune cells in the digestive tract?

Q4. In Figure 4, the DC cells seem to be in direct contact with gut microbiota, which might be contradictory with the statement of “The chemical and physical barrier of the intestine keeps the gut microbiota and immune cells apart, preventing abnormal immune reactions and allowing them to co-exist in harmony within the host”.

Q5. In “7. Human and Animal Trials on Probiotics” section, the expression of “Probiotics have been proven beneficial in alleviating symptoms of insulin resistance” lacks relevant references and mechanisms.

Q6. Many of the statements in the text are ambiguous. Such as in “8. Role of Probiotics in Food Allergy” section, if the number of bacteria that are associated with food allergy is stated only at the family level, the reader still does not know which identified bacteria should be supplemented to prevent or treat food allergy.

This review exhibited sufficient content, the probiotics and health benefits, but many statements are suggested for further improvements, and the repetition rate is too high.

Author Response

Dear reviewer 2,

Thank you for taking the time to review our paper and for your thoughtful and constructive comments. Please find our response to your comments and suggestions in the attached pdf-file.

Regards

Ralf Weiskirchen

Reviewer 3 Report

Comments and Suggestions for Authors

In the present work, Chandrasekaran try to review the effects of probiotics on gut microbiota. It is known that probiotics can regulate intestinal flora to enhance host immunity. In this manuscript, the latest findings on gut microbiota and their influence on human health, as well as the effects of probiotics on the composition and function of the human gut microbiota, and host immunity in building a healthy intestine are discussed. In addition, the implications of probiotics in some of the most important human diseases are reviewed. However, there are some questions that should be explained. Major concerns 1. Figure 1, it is showed that the colony-forming units in the human gastrointestinal tract increases from the stomach/duodenum to the jejunum/ileum to the colon. It is well known that the small intestine has three distinct regions: the duodenum, jejunum, and ileum, and the large intestine includes the cecum, colon, rectum, and anus. The colony-forming units in the cecum and rectum are not showed. In addition, arrow for the duodenum should be move down. 2. Figure 2, the centre of figure should be showed only gut, and liver and stomach should be moved. In addition, using a thyroid picture for the endocrine system is not suitable. 3. Figure 3, the structure of gut is not right, which should be revised. In general, the structure of gut is comprised of mucosa (simple columnar epithelial tissue), submucosa (blood vessels, nerves and connective tissue), muscularis (the muscularis) and serosa (simple squamous epithelial tissue. 4. Figure 4, the intestinal cavity and the structure of gut, including mucosa, submucosa, muscularis and serosa, should be indicated. Minor concerns 1. A summary sentence may be added in the end of the Abstract section. 2. Lines 52-54, a reference should be added. 3. Lines 59-65, a reference should be added. 4. Lines 162-162, too short paragraph. 5. Line 238, ‘Furthermore,’ at the start of a paragraph is not suitable. 6. Figure 5, the words (XXX-Axis) should be moved nearby to the arrows. 7. The format of reference is not suitable for this Journal.

Comments on the Quality of English Language

Moderate editing of English language required.

Author Response

Dear reviewer 3,

Thank you for taking the time to review our paper and for your thoughtful and constructive comments. Please find our response to your comments and suggestions in the attached pdf-file.

Regards

Ralf Weiskirchen

Round 2

Reviewer 2 Report

Comments and Suggestions for Authors

Thanks  for these modifications, while the amount of wording duplication in the manuscript is still too high. Generally, less than 30% is considered acceptable.

Author Response

Dear Reviewer 2,   many thanks for sending screeing the similarity. We have now tried to reduce the similarity index. I hope your screen will show a significant reduction.   Regards Ralf Weiskirchen

Reviewer 3 Report

Comments and Suggestions for Authors

Thanks for author’s responses. However, the percent match is 40% for this manuscript in the iThenticate report, which should be reduced.

Comments on the Quality of English Language

Minor editing of English language required.

Author Response

Dear Reviewer 3,   many thanks for sending screeing the similarity. We have now tried to reduce the similarity index. I hope your iThenticate screen will show a significant reduction.   Regards Ralf Weiskirchen